# Synthesis and Characterization of Colistin-Functionalized Silica Materials for Rapid Capture of Bacteria in Water

**DOI:** 10.3390/molecules27238292

**Published:** 2022-11-28

**Authors:** Jingli Qiu, Jianli Li, Xiaoxi Du, Tong Zhou, Bingbing Xie, Limin He

**Affiliations:** 1Guangdong Provincial Key Laboratory of Veterinary Pharmaceutics Development and Safety Evaluation, South China Agricultural University, Guangzhou 510642, China; 2Guangdong Laboratory for Lingnan Modern Agriculture, Guangzhou 510642, China; 3National Reference Laboratory of Veterinary Drug Residues (SCAU), College of Veterinary Medicine, South China Agricultural University, Guangzhou 510642, China; 4Quality Supervision, Inspection and Testing Center for Domestic Animal Products Guangzhou, Ministry of Agriculture and Rural Affairs, Guangzhou 510642, China

**Keywords:** colistin modifier, SiO_2_@NH_2_@COOH@CST, characterization, rapid capture, bacteria, water

## Abstract

In this study, a new colistin-functionalized silica gel material (SiO_2_@NH_2_@COOH@CST) was synthesized after carboxylation on the surface of amino-modified silica. The main factors affecting the adsorptive properties of the material, such as the types of linkers, the linking methods, the reaction buffers and the particle sizes of carriers, were systematically investigated. The SiO_2_@NH_2_@COOH@CST was characterized by means of electron microscopy, Fourier-transform infrared spectroscopy, zeta potential measurements, etc. We demonstrated that the sorbent showed good adsorption of Gram-negative bacteria. The adsorption efficiency of *E*. *coli* on SiO_2_@NH_2_@COOH@CST was 5.2 × 10^11^ CFU/g, which was 3.5 times higher than that on SiO_2_@NH_2_@COOH, suggesting that electrostatic interactions between SiO_2_@NH_2_@COOH@CST and *E*. *coli* played a key role. The adsorption was quick, and was reached in 5 min. Both pseudo-first-order and pseudo-second-order kinetic models fit well with the dynamic adsorption process of SiO_2_@NH_2_@COOH@CST, indicating that physical adsorption and chemisorption might occur simultaneously during the adsorption process. SiO_2_@NH_2_@COOH@CST was successfully applied for the rapid capture of bacteria from water. The synthesized material could be used as a potential means of bacterial isolation and detection.

## 1. Introduction

Colistin (CST), also known as polymyxin E, is one of the polymyxin antibiotics. Nowadays, the irregular use of antibiotics has led to the production of multidrug-resistant (MDR) bacteria. Owing to the emergence of MDR strains, traditional antibiotics have become increasingly ineffective [1,2]. Not only can CST prevent diseases and promote growth in global animal husbandry production; it also has the ability to be effective against MDR bacterial infections, including most Gram-negative bacteria (GNB), such as *Escherichia coli* (*E*. *coli*), *Klebsiella*, *Salmonella*, *Acinetobacter baumannii* (*A*. *baumannii*), and so on [3]. CST is characterized by a positively charged cationic hydrophilic fragment and a hydrophobic chain structure, which is derived from 2,4-diaminobutyric acid (DAB) units and a hydrophobic fatty acyl tail, respectively. The main target site of CST against GNB is lipopolysaccharide (LPS, endotoxin) on the outer membrane [4]. The LPS structure is composed of a lipid A moiety, a conserved oligosaccharide core (2-keto-3-deoxyoctonoic acid), and an O-antigen group. Lipid A carries a large amount of negative charge due to the existence of phosphate groups [5]. Primarily, CST and lipid A demonstrates electrostatic adsorption through the specific combination of the positively charged free amino groups of DAB residues and the negatively charged phosphate groups. Thereafter, the hydrophobic domain, including an N-terminal fatty acyl chain and a hydrophobic part, is inserted into the outer membrane of GNB. CST exerted antimicrobial activity, resulting in the expansion of the outer membrane of the bacteria as well as an imbalance in osmotic pressure, and ultimately the bacteria died [6]. The cell wall of most Gram-positive bacteria (GPB) lacks an LPS-containing outer membrane, which is different from that of GNB. Its cell wall is principally constituted of teichoic acids and a thick peptidoglycan layer [7].

The distinctive bactericidal mechanism of CST is based on electrostatic interaction, which implies that CST could be immobilized on a suitable carrier, which could in turn capture bacteria as an adsorbent. In practice, as early as 1994 it was reported that polymyxin B (PMB) has been immobilized on fibers to selectively remove endotoxin from the blood [8]. Moreover, PMB was also immobilized on cross-linked cellulose microspheres [9] and quartz [10] for endotoxin adsorption. Recently, the literature has shown an accumulation of interest regarding whether immobilized CST on a silica carrier could capture bacteria through the electrostatic interactions with the LPS part on its outer membrane.

Water is one of the most important microbial habitats on earth, particularly for GNB [11]. It has been reported that low infectious doses of pathogenic bacteria in water not only pose a serious threat to human health, but also cause widespread damage [12]. The main reason that GNB is a public-health challenge is the presence of an outer cell-protecting membrane, which plays an important role in the transfer of antibiotic genes [13]. *E*. *coli* is the most commonly used fecal indicator in water quality monitoring and water safety management. It can cause diarrhea, hemorrhagic colitis, and hemolytic uremia syndrome [2]. Currently, disinfectants such as chlorine, chloramines, and ozone are commonly used to eliminate pathogens from water. However, these chemicals react with various components in natural water to produce more than 600 disinfection by-products (DBPs), many of which are carcinogenic and potentially harmful to the human body [14]. As a result, the development of new bacterial decontamination and extraction strategies is of great urgency and importance.

Functional materials applied to remove bacteria from environmental water could be an alternative to traditional disinfection and sterilization methods. The relevant literature has reported on modifying the surfaces of magnetic particles with cationic groups [15], antibodies [16], antibiotics [17], and other groups [18,19] for the removal or detection of bacteria. Nevertheless, the synthesis of adsorption materials with CST as a functional agent to capture bacteria has rarely been studied. Carrión et al. synthesized CST-functionalized CdSe/ZnS quantum dots as a fluorescent probe for the rapid detection of *E*. *coli* [20]. Bell et al. extracted *A. baumannii* by preparing CST-functionalized γFe_2_O_3_/Au core/shell magnetic nanoclusters [21]. These different magnetic nanoparticles were used to extract and detect bacteria from blood [17], drinking water [22], chicken meat supernatants [23], lake water, urine, etc. [24].

In this study, micron-scale bare silica (SiO_2_) particles were first amino-modified and then carboxylated; finally, CST, as a functional modifier, was covalently bonded to the surface of carboxyl-activated silica gels. The key factors, such as the linkers used, linking modes, and silica particle size, were investigated during the synthesis process. The adsorption models and mechanisms of the CST-functionalized silica (SiO_2_@NH_2_@COOH@CST) were analyzed, and the sorbent was used for the rapid capture of *E*. *coli* from water. This new material represents a potential approach for bacterial isolation and detection.

## 2. Results and Discussion

### 2.1. Preparation of SiO_2_@NH_2_@COOH@CST

#### 2.1.1. Selection of Synthesis Route

Inspired by the physical bactericidal mechanism based on the unique chemical structure of CST, we guessed that CST may represent a good bacterial capture agent. In this study, the CST was immobilized on the silica carrier for the selective adsorption of bacteria.

There are several approaches available for the grafting of polymyxins onto different carriers. Cao et al. designed and prepared a PMB-immobilized cross-linked cellulose microsphere with potential applications in the field of blood purification. First, the microspheres were modified with 1,4-butanediol diglycidyl ether to expose epoxyl groups on the surface of the cellulose microspheres, and then PMB was linked to the microspheres using epoxyl groups [9]. CST molecules were attached to the terminal carboxyl group of mercaptoacetic-acid-capped quantum dots in the presence of 1-(3-Dimethylaminopropyl)-3-ethylcarbodiimide hydrochloride (EDC·HCl) and N-Hydroxy succinimide (NHS) as amide bond promoters. The potential application of these CST-modified nanoparticles for the sensitive detection of *E*. *coli* bacterial cells was demonstrated [21]. In another study, CST was conjugated to ortho-pyridine disulfide and an N-hydroxysuccinimide heterobifunctional polyethylene glycol polymer linker, and then superparamagnetic FeOx/Au core/shell nanoparticles were functionalized with the CST-modified conjugates. The final magnetic nanoclusters were used to magnetically capture and extract *A. baumannii* using CST as the microbial targeting ligand [22]. In this study, we selected micron-scale bare silica as a carrier. SiO_2_ was first amino-modified using 3-amino-propyltriethoxysilane (APTES), and then carboxylation on SiO_2_@NH_2_ was carried out using succinic anhydride (SA). The carboxyl groups on the surface of carboxyl-modified silica gel (SiO_2_@NH_2_@COOH) were activated in the presence of EDC·HCl /NHS, and further reacted with the NH_2_ groups of CST. Thereby, CST was covalently grafted onto SiO_2_@NH_2_@COOH, and the final CST-functionalized silica particles (SiO_2_@NH_2_@COOH@CST) were obtained.

#### 2.1.2. Optimization of Linkers

The effect of grafting CST via linkers with different bridge groups on bacteria adsorption was evaluated (synthetic process shown in Appendix A). The surface of SiO_2_ was modified via CST using several linkers and linking methods, including APTES-SA with (or without) 6-Aminocaproic acid (6-AA), APTES- hexamethylene diisocyanate (HDMI)-6-AA, and 3-Isocyanatopropyltriethoxysilane (IPTS)-6-AA. As shown in Figure 1, the adsorption capacity of the material synthesized using IPTS and APTES-HDMI as linking agents was lower than that of using APTES-SA, and the adsorption values were 1.4, 2.9, and 4.7 × 10^11^ CFU/g, respectively. This was probably due to the lower grafting ratio of CST when IPTS and HDMI were used as linking agents. In addition, the adsorption capacity of the material synthesized with or without 6-AA as the linking agent was higher. The addition of 6-AA did not play a role, despite the fact that it may add some extra flexibility [25].

#### 2.1.3. Selection of Buffer Systems

Three kinds of reaction buffer solutions, including boric acid buffer, phosphate buffer, and 2-(N-Morpholino)-ethanesulfonic acid monohydrate (MES•H_2_O) buffer (all at a concentration of 50 mM and pH 5.5), were assessed. The SiO_2_@NH_2_@COOH@CST synthesized in MES buffer exhibited the best adsorption capacity, and the adsorption capacity reached 3.3 × 10^11^ CFU/g. The adsorption capacities of SiO_2_@NH_2_@COOH@CST synthesized in boric acid buffer and phosphate buffer were 2.3 and 1.9 × 10^11^ CFU/g, respectively (Appendix A). The results suggested that the SiO_2_@NH_2_@COOH@CST synthesized in the organic (MES) buffer system had a stronger adsorption capacity than that synthesized in the inorganic (boric acid and phosphate) buffer system. Accordingly, EDC·HCl was more unstable in PBS than in the MES buffer [26]. Therefore, the MES buffer was selected for the subsequent experiments.

#### 2.1.4. Optimization of Silica Particle Size

Silica particle diameters of 2.7, 3.7, 5, and 50 µm were selected to investigate the effect of particle size on bacterial adsorption. As shown in Figure 2, the adsorption capacities of SiO_2_@NH_2_@COOH@CST materials prepared with different-sized particles as carriers was markedly different. With an increase in the silica particle diameter from 2.7 to 50 µm, the adsorption capacity of the CST-modified material in regards to *E*. *coli* gradually decreased from 4.0 to 1.3 × 10^11^ CFU/g. The adsorption capacity of SiO_2_@NH_2_@COOH@CST synthesized using 2.7 µm silica particles was about three times higher than that synthesized using 50 µm particles.

In the literature, we identified a few reports on adsorbing bacteria in which differences in particle size further affected the bacterial adsorption. Knowles et al. demonstrated that the numbers of bacteria that adhered to the larger 30 and 75 nm nanoparticle surfaces were significantly lower than those on the surfaces of 7 and 12 nm particles [27]. It has been reported that the amount of *Serratia marcescens* cells attached to quartz particles increased as the particle size decreased from 1000 µm to <1.5 µm [28]. Wu et al. investigated the adsorption of *P. putida* on different-sized particles of soil and demonstrated that the maximum amount of *P. putida* adsorbed on the clay (<2 µm) fraction was 4.3 and 62.3 times larger than those on the silt (2–20 µm) and sand (20–2000 µm) fractions, respectively [29]. As is well known, the smaller the particle size, the larger the specific surface area [30]. Therefore, when using silica with different-sized particles as carriers, with an increase in the specific surface area, the grafting amount of CST would increase, in addition to an increase in the interactions between bacteria and CST-modified silica particles, thereby improving the adsorption capacity of SiO_2_@NH_2_@COOH@CST in relation to bacteria.

### 2.2. Characterization

#### 2.2.1. Scanning Electron Microscopy (SEM) and Transmission Electron Microscopy (TEM) Analysis

The images of SiO_2_, SiO_2_@NH_2_@COOH, and SiO_2_@NH_2_@COOH@CST were compared using SEM and TEM. As shown in Figure 3, SiO_2_, SiO_2_@NH_2_@COOH, and SiO_2_@NH_2_@COOH@CST were all regular spheres with a smooth surface and a uniform shape and were relatively dispersed, indicating that modification and grafting reactions did not cause the silica to agglomerate and break. Since the molecular weight of CST grafted on the surface of silica was small, it was not sufficient to cause a significant change. The morphology of the three kinds of silica gels was further characterized via TEM. The three kinds of silica gels were all dispersed and had a smooth surface, which was consistent with the SEM results (Figure 3). Energy dispersive spectroscopy (EDS) of the TEM confirmed the presence of C, N, Si, and O elements in the SiO_2_@NH_2_@COOH@CST based on the presence of their characteristic X-rays. The copper (Cu) TEM grid contributed to the elemental presence of Cu in the EDS spectra (Appendix A).

#### 2.2.2. Fourier-Transform Infrared Spectroscopy (FT-IR) Analysis

In order to confirm the reaction process, the FT-IR spectra of the prepared SiO_2_@NH_2_, SiO_2_@NH_2_@COOH, and SiO_2_@NH_2_@COOH@CST were tested (Appendix A). For all three silica gels, the same absorption bands at 2964 cm^−1^ and 2938 cm^−1^ were observed, which were mainly characteristic stretching vibrations of the CH_2_ group. The characteristic bands around 3441 cm^−1^ and 1630 cm^−1^ were vibrations of N-H bonds. Compared to SiO_2_@NH_2_, strong vibrations appeared at 1699 cm^−1^, 1654 cm^−1^, and 1559 cm^−1^ in SiO_2_@NH_2_@COOH. The absorption at 1699 cm^−1^ was probably caused by the vibrations of C=O in the carboxyl group. The peaks at 1654 cm^−1^ and 1559 cm^−1^ possibly resulted from the C=O and N-H in the amide bond, which was generated by the reaction between the amino group on the SiO_2_@NH_2_ and SA. Compared with SiO_2_@NH_2_@COOH, the absorption peak of SiO_2_@NH_2_@COOH@CST at 1699 cm^−1^ was weakened but had not completely disappeared, whereas the characteristic absorption peaks of the amide bond at 1654 cm^−1^ and 1559 cm^−1^ were obviously enhanced. We speculated that the peak at 1699 cm^−1^ was the characteristic absorption peak of C=O in the carboxyl group, and those at 1654 cm^−1^ and 1559 cm^−1^ were the stretching vibration peaks of C=O and N-H in CST, indicating that CST was successfully covalently grafted onto the surface of SiO_2_@NH_2_@COOH.

#### 2.2.3. Thermogravimetric Analysis

Thermogravimetric analysis was performed to evaluate the relative content of CST grafted onto the surface of SiO_2_@NH_2_@COOH. The weight loss curves of the two silica gels were similar, and there was no obvious change up to 100 °C. The curves began to decrease when the temperature rose to 200 °C. The mass losses of SiO_2_@NH_2_@COOH and SiO_2_@NH_2_@COOH@CST were 20.6% and 22.2% at 900 °C, respectively (Appendix A). The results regarding weight loss suggested that the mass loss of SiO_2_@NH_2_@COOH@CST was increased by 1.6% relative to SiO_2_@NH_2_@COOH.

#### 2.2.4. Zeta Potential Measurement

The surface charges of SiO_2_, SiO_2_@NH_2_, SiO_2_@NH_2_@COOH, and SiO_2_@NH_2_@COOH@CST were analyzed through the use of a zeta potentiometer. It was shown that with a change in the surface-modified groups of the silica, the surface potential was quite different. The zeta potential of bare SiO_2_ was 3.3 ± 0.5 mV, whereas the surface potential of SiO_2_@NH_2_ increased to 18.1 ± 2.4 mV, indicating that the amino group was successfully connected to the surface of SiO_2_. After the surface of SiO_2_@NH_2_ was further modified by SA, there were large carboxyl groups on the surface of SiO_2_@NH_2_; therefore, the zeta potential of SiO_2_@NH_2_@COOH became a negative value, at −23.6 ± 0.3 mV. Finally, the potential of SiO_2_@NH_2_@COOH@CST changed to a positive value, at 20 ± 1.1 mV, probably owing to the cationic characteristics of CST. These results hinted that the designed modifiers were successfully linked to the silica surface.

### 2.3. Adsorption Properties

#### 2.3.1. Adsorption Isotherm

We assessed the relationship between the concentration of *E*. *coli* and modified silica gels at room temperature using an adsorption isotherm. As shown in Figure 4, with an increase in the initial substrate concentration, the adsorption capacity of SiO_2_@NH_2_@COOH did not rise noticeably. However, that of SiO_2_@NH_2_@COOH@CST increased gradually and reached adsorption saturation at 5.08 × 10^11^ CFU/g when the initial *E*. *coli* concentration was at 6 × 10^9^ CFU/mL. Notably, the adsorption capacity of SiO_2_@NH_2_@COOH@CST was always higher than that of SiO_2_@NH_2_@COOH (about 3.5 times in adsorption saturation). The difference in adsorption capability between SiO_2_@NH_2_@COOH and SiO_2_@NH_2_@COOH@CST might be due to the fact that under physiological pH conditions, the surface of the former was mainly negatively charged, which repelled negatively charged *E*. *coli* on the surface, whereas the surface of the latter was positively charged, which contributed to capturing *E*. *coli*. The electrostatic interactions between SiO_2_@NH_2_@COOH@CST and *E*. *coli* play a key role in the adsorption process [6].

#### 2.3.2. Adsorption Kinetics

Adsorption kinetic experiments are commonly used to study the adsorption properties of sorbents. As shown in Figure 5, the capture rates of *E*. *coli* from aqueous solution by SiO_2_@NH_2_@COOH@CST and SiO_2_@NH_2_@COOH were rapid, and the adsorption equilibrium was reached within the first 5 min. There was no significant increase in adsorption after 5 min of incubation. The rapid adsorption rate might be mainly ascribed to the thin grafting coating layer on the surface of silica and the large specific surface area of the carrier, as well as the high concentration of bacteria, which considerably decreased the mass transfer resistance. Carrillo-Carrión et al. developed an approach for screening bacteria using CST-functionalized CdSe/ZnS nanoparticles, which was completed in only 15 min as well [20]. Miller et al. synthesized CST-functionalized gold nanoparticles, which could be rapidly attached to *A. baumannii*, and achieved half-maximum saturation in around 7 min [31]. In this study, although SiO_2_@NH_2_@COOH@CST and SiO_2_@NH_2_@COOH had similar dynamic adsorption rates, the adsorption capacity of SiO_2_@NH_2_@COOH@CST was far higher than that of SiO_2_@NH_2_@COOH over the entire time of the experiment, which indicated that the grafting of CST onto SiO_2_@NH_2_@COOH was the main reason for the effective adsorption of *E*. *coli*.

In terms of SiO_2_@NH_2_@COOH@CST, the correlation coefficient (r^2^), Qe, and k values of the pseudo-first-order and pseudo-second-order kinetic models were 0.975, 3.42 × 10^11^ CFU/g and 0.591 min^−1^, and 0.976, 3.46 × 10^11^ CFU/g, and 0.749 g × 10^11^ CFU^−1^ min^−1^, respectively. The pseudo-first-order kinetic model was primarily applied to physical adsorption, whereas the pseudo-second-order kinetic model was primarily applied to chemisorption [32]. The r^2^ value of the two models was greater than 0.9 and similar, suggesting that the adsorption process could be accounted for in both models [33,34], which could be considered to be a consequence of the synchronous occurrence of physical adsorption and chemical adsorption. The NH_2_ group carried on CST and the negatively charged phosphate group on LPS can generate electrostatic interaction [6]. Meanwhile, SiO_2_@NH_2_@COOH@CST might also chemically react with other groups on the surface of bacteria, resulting in chemical adsorption. Consequently, the above results indicated that the adsorption process between *E*. *coli* and SiO_2_@NH_2_@COOH@CST was complicated, and further research is needed to clarify this mechanism.

With regard to SiO_2_@NH_2_@COOH, the r^2^, Qe, and k values of the pseudo-first-order and pseudo-second-order models were 0.857, 1.04 × 10^11^ CFU/g, and 0.571 min^−1^ and 0.855, 1.05 × 10^11^ CFU/g, and 4.27 g × 10^11^ CFU^−1^ min^−1^, respectively. The SiO_2_@NH_2_@COOH control samples, in the absence of CST, did not present strong correlations in terms of their binding behavior with *E*. *coli* when fitting the above models. This further confirmed that the silica gel’s attachment to *E*. *coli* was dependent upon the presence of the CST targeting ligand. The rapid rate of silica gel-bacteria binding mediated by CST was considered to be a result of new bacterial removal and isolation methods.

### 2.4. Factors Affecting Bacterial Adsorption

#### 2.4.1. Effect of pH on *E. coli* Adsorption

In this study, the effects of different pH values (3, 5, 7, and 9) on *E*. *coli* adsorption by the new material were investigated. When the pH of the bacterial suspension was increased from 3 to 9, the efficiency of the adsorption of *E*. *coli* by SiO_2_@NH_2_@COOH@CST decreased sharply. At pH 3, its adsorption efficiency reached its highest point, at 7.57 × 10^11^ CFU/g. At pH 5, 7, and 9, the adsorption efficiency of SiO_2_@NH_2_@COOH@CST decreased by 3.06%, 55.5%, and 86.5%, respectively, indicating that the adsorption efficiency of SiO_2_@NH_2_@COOH@CST to *E*. *coli* is significantly affected by the pH of the bacterial suspension.

Bacterial capture is influenced by the surface charge, hydrophobicity, and surface properties of the adsorbent. The surface of *E*. *coli* possesses a purely negative charge by virtue of the ionized phosphoryl substituents on outer cell envelope macromolecules [15]. We speculated that the influence of pH on the adsorption efficiency of *E*. *coli* was mainly due to the changes in the surface charge of the adsorbent and *E*. *coli* with the change in pH. The electrostatic interaction of SiO_2_@NH_2_@COOH@CST and *E*. *coli* in solution was enhanced at low pH levels, in which the SiO_2_@NH_2_@COOH@CST surface was positively charged due to protonation. With the increase in pH, the degree of protonation on the surface of the adsorbent was significantly reduced, which weakened the electrostatic effect and resulted in a sharp decrease in the adsorption capacity of *E*. *coli* [35]. Consequently, with an increase in pH of the bacterial suspension, the adhesive attraction between SiO_2_@NH_2_@COOH@CST and *E*. *coli* would gradually decrease, and would even be converted to electrostatic repulsion. These results suggest that electrostatic interactions played an important role in the adsorption process.

#### 2.4.2. Effect of Temperature on *E. coli* Adsorption

Generally, temperature has an important influence on the properties of adsorbents. Herein, three different temperatures—4 °C, 25 °C, and 37 °C were selected to evaluate the effect of temperature on *E*. *coli* adsorption. The results showed that the adsorption capacity increased gradually with the increase in temperature from 4 °C to 37 °C; the adsorption contents of *E*. *coli* were 2.23, 2.77, and 3.69 × 10^11^ CFU/g, respectively. Berkeley et al. pointed out that the degree of bacterial adsorption on a solid surface was affected by the bacterial physiological state, and vigorous bacterial metabolism promoted the adsorption capacity [36]. As is well known, the optimal growth temperature of *E*. *coli* is 37 °C, which might be the reason why the adsorption efficiency of *E*. *coli* was the highest in this study.

### 2.5. Application

The feasibility of SiO_2_@NH_2_@COOH@CST was investigated through the capturing of bacteria from real water samples. The results are shown in Figure 6. For all of the bacteria including GNB and GPB, the SiO_2_@NH_2_@COOH@CST exhibited great adsorption capacity in tap water, lake water, and Pearl River water, but poor adsorption capacity in farm water. In addition to capturing most of the GNB, the SiO_2_@NH_2_@COOH@CST was also able to capture GPB such as *S. aureus*, due to a large amount of teichoic acids on their surface, which can generate electrostatic interactions with the amino groups on CST [7]. The adsorption differences observed for bacteria in different environmental water sources might be related to the complex matrices in the water bodies, which affect the adsorption capacity of materials [37]. Furthermore, we speculated that the different pH values of the tap water, lake water, Pearl River water, and farm water also affected the adsorption. We intend to carry out further research on this issue.

## 3. Materials and Methods

### 3.1. Reagents and Materials

EDC·HCl, NHS, MES•H_2_O, SA, and 6-AA were supplied by the Macklin Company (Shanghai, China). IPTS and HDMI were available from the Energy Chemical Company (Shanghai, China) and the J&K Scientific Company (Beijing, China). APTES was provided by Sigma Aldrich (Shanghai, China) Trading Co Ltd. Other analytical-grade solvents including N, N-dimethylformamide (DMF) and methanol (MeOH) were obtained from the Guangzhou Chemical Reagent Factory (Guangzhou, China). Ultrapure water was produced using a Milli-Q water system (Molsheim, France). Silica (50 µm) was obtained from Silicycle (Quebec, Canada). High-purity silica (SPS100-5 µm, SPS100-3.7 µm, SPS100-2.7 µm) was purchased from FUJI Corporation (Aichi, Japan). CST Sulfate was purchased from Newprobe (Beijing, China).

### 3.2. Preparation of CST-Modified Silica

#### 3.2.1. Synthesis of Amino-Modified Silica (SiO_2_@NH_2_)

Five milliliters of APTES and 1 mL triethylamine were added to a mixture solution with 100 mL of anhydrous toluene and 5 g of silica in a 250 mL round-bottom flask. The reaction was performed under a nitrogen (N_2_) atmosphere at 120 °C for 12 h. The SiO_2_@NH_2_ particles were collected using a centrifuge at 6000 rpm for 5 min and washed with MeOH and DMF several times to remove residual APTES. Finally, SiO_2_@NH_2_ was dried at 50 °C for 24 h in a vacuum chamber.

#### 3.2.2. Carboxylation on the Surface of SiO_2_@NH_2_

Two grams of SiO_2_@NH_2_ and 4 g of SA were added into DMF. The mixture was stirred at 120 °C overnight under an N_2_ atmosphere. The resultant silica was washed with DMF, MeOH, and water several times to remove excess reactants and byproducts. Finally, the synthesized SiO_2_@NH_2_@COOH was dried at 50 °C for 24 h in a vacuum chamber.

#### 3.2.3. Synthesis of SiO_2_@NH_2_@COOH@CST

One gram of SiO_2_@NH_2_@COOH, 50 mL of MES buffer (50 mM, pH 5.5 ± 0.1), and 100 mg of EDC·HCl and NHS were added into the 100 mL flask and reacted at room temperature for 1 h. Then, the activated silica was washed with MES buffer 2 times and then added into the 50 mL MES buffer solution containing 900 mg CST. The mixture was vortexed and allowed to react at room temperature for 2 h. After reaction, SiO_2_@NH_2_@COOH@CST was washed with MEOH and water several times to remove unreacted CST. Finally, the product, SiO_2_@NH_2_@COOH@CST, was dried at 50 °C for 24 h in a vacuum chamber before storage.

### 3.3. Characterization of SiO_2_@NH_2_@COOH@CST

A ZEISS EVO MA 15 microscope (Oberkochen, Germany) and a FEI Talos L120C system (Waltham, MA, USA) were used to obtain the SEM and TEM images of different silica particles. FT-IR was performed using a Bruker Vertex 70 infrared spectrometer (Bruker, Germany), recording from 4000 cm^−1^ to 500 cm^−1^. The dried samples were mixed with KBr and detected. The thermostability of silica particles was determined using a PerkinElmer STA 6000 thermo gravimetric analyzer (Waltham, MA, USA). Analysis was performed at the following parameters: nitrogen pressure, 0.2 MPa; initial temperature, 50 °C; final temperature, 900 °C; heating rate, 20 °C/min. Zeta potentials of different silica particles’ surfaces were measured using a Malvern Zetasizer Nano ZS 90 (Malvern, UK).

### 3.4. Preparation of Bacteria Suspension

The capacity of SiO_2_@NH_2_@COOH@CST to capture GNB was evaluated by using *E*. *coli* ATCC 25922 as a model microorganism. Frozen colonies of bacteria were initially inoculated into Luria–Bertani (LB) medium and then cultivated at 37 °C for 10 h in a rotary shaker. The bacteria were harvested via centrifugation at a stationary growth phase at 4000 rpm for 10 min and washed two additional times with sterilized physiological saline solution (0.9% NaCl) to remove the traces of the growth medium. The bacterial mud was subsequently resuspended in the physiological saline solution for adsorption experiments.

### 3.5. Adsorption Experiments

#### 3.5.1. Measurement of the Adsorbed Bacteria

Ten milligrams of modified silica particles were added into 5 mL of *E*. *coli* bacterial solutions (2~4 × 10^9^ CFU/mL). The mixture solution was shaken at 150 rpm for 10 min. Then, 2 mL of sucrose solution (60% by weight) was added into the suspension, which was able to create a density gradient to separate the bacteria that were adsorbed by the silica gel and the bacteria that were not adsorbed [38]. The free bacteria were separated from those adsorbed by silica particles via centrifuging the mixture at 2000 rpm for 20 min. The silica-cell aggregates sank to the bottom of the tube and the bacteria not bound to silica particles were still in the suspension, which was determined via wet-weight measurements [39]. The wet weight of the adsorbed bacteria was calculated based on the difference between the total wet weight of bacteria and the wet weight of bacteria recovered in the suspension. All experiments were analyzed in triplicate and data were presented as the means ± standard deviations (SD).

#### 3.5.2. Adsorption Isotherm and Kinetic Adsorption

An adsorption isotherm was implemented by mixing 10 mg of modified silica particles and 5 mL of *E*. *coli* in various total wet-weight concentrations (6, 8, 20, 40, 60, and 80 × 10^8^ CFU/mL) at room temperature. In the study of the adsorption isotherms, we referred to the concentration relationship between *E*. *coli* and the adsorbents. Kinetic adsorption analysis was carried out by mixing 10 mg of SiO_2_@NH_2_@COOH@CST and 5 mL of 4 × 10^9^ CFU/mL *E*. *coli* for different times, that is, 5, 15, 30, 45, 60, and 120 min.

### 3.6. Application

The SiO_2_@NH_2_@COOH@CST prepared in this study was applied to the capture of bacteria spiked around 5 mg/mL in real water samples, including tap water, lake water, Pearl River water, and farm water. The water samples were rested overnight and vacuum-filtered through 0.45 µm microporous membranes to remove fine particles and bacteria. The adsorption efficiency of the modified silica particles was tested on mild bacteria such as *E*. *coli* (E55), *P*. *aeruginosa* (P10), *S*. *aureus* (S2, GPB), and *Salmonellae* (S1, ATCC14028) isolated from local chicken farms.

## 4. Conclusions

In this study, we synthesized and characterized a new CST-functionalized silica material, SiO_2_@NH_2_@COOH@CST. The prepared material, used as a sorbent, can effectively capture bacteria. Electrostatic interactions play a key role in the capture of bacteria by SiO_2_@NH_2_@COOH@CST. The CST-functionalized sorbent can represent a potential means to quickly remove and isolate bacteria from environmental water.

## Figures and Tables

**Figure 1 molecules-27-08292-f001:**
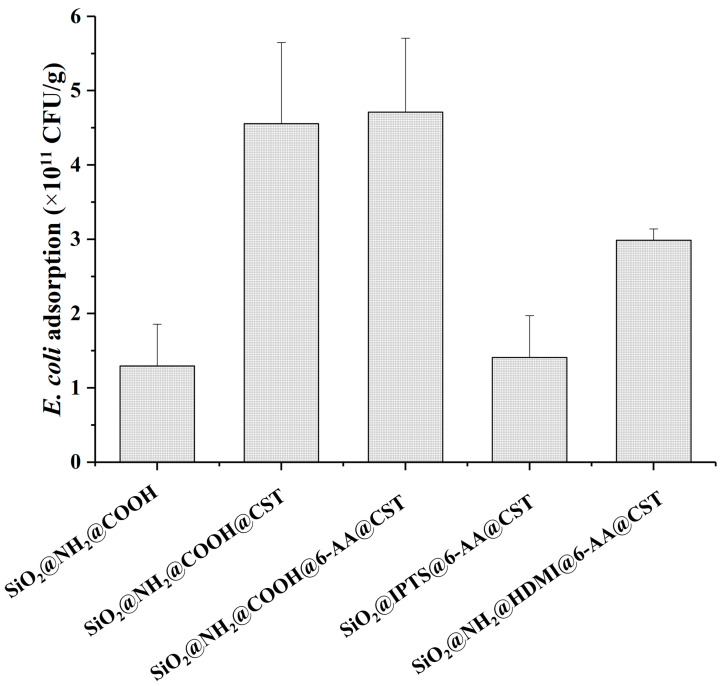
Effect of linkers and linking modes on the adsorption of *E*. *coli* by different modified silica gel.

**Figure 2 molecules-27-08292-f002:**
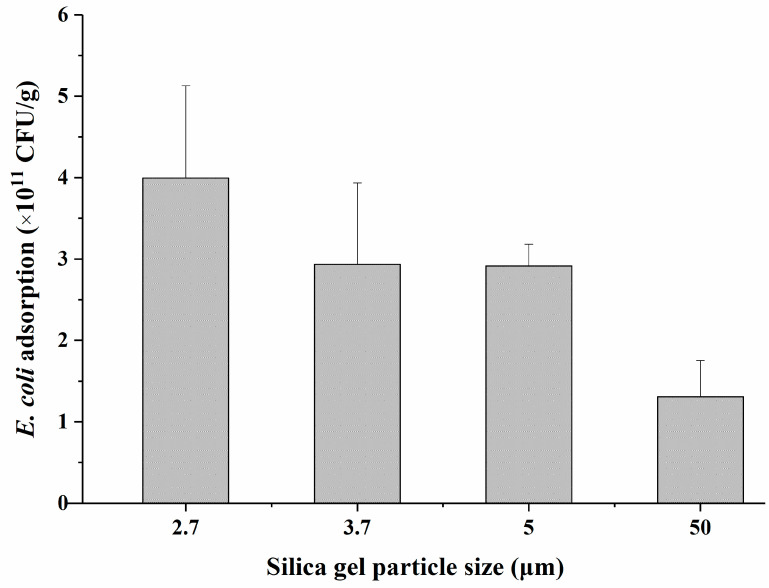
Influence of the use of different particle sizes on the adsorption of *E*. *coli* by SiO_2_@NH_2_@COOH@CST.

**Figure 3 molecules-27-08292-f003:**
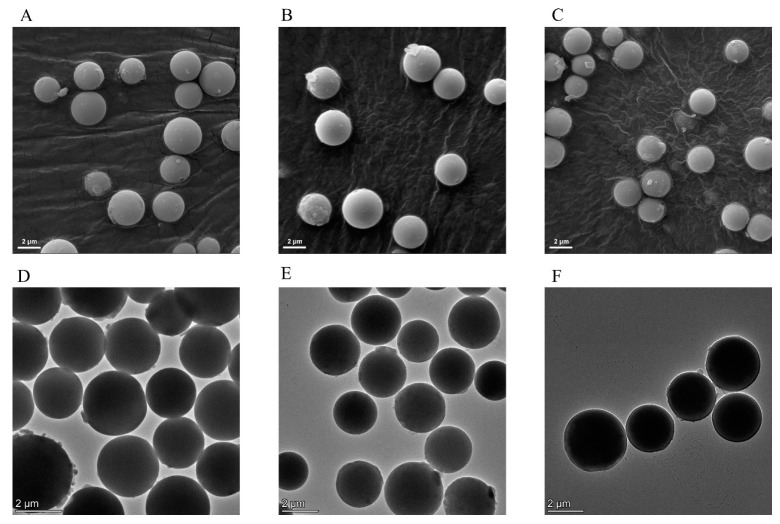
SEM micrographs of bare active silica gel (**A**), SiO_2_@NH_2_@COOH (**B**), and SiO_2_@NH_2_@COOH@CST (**C**). TEM micrographs of bare active silica gel (**D**), SiO_2_@NH_2_@COOH (**E**), and SiO_2_@NH_2_@COOH@CST (**F**).

**Figure 4 molecules-27-08292-f004:**
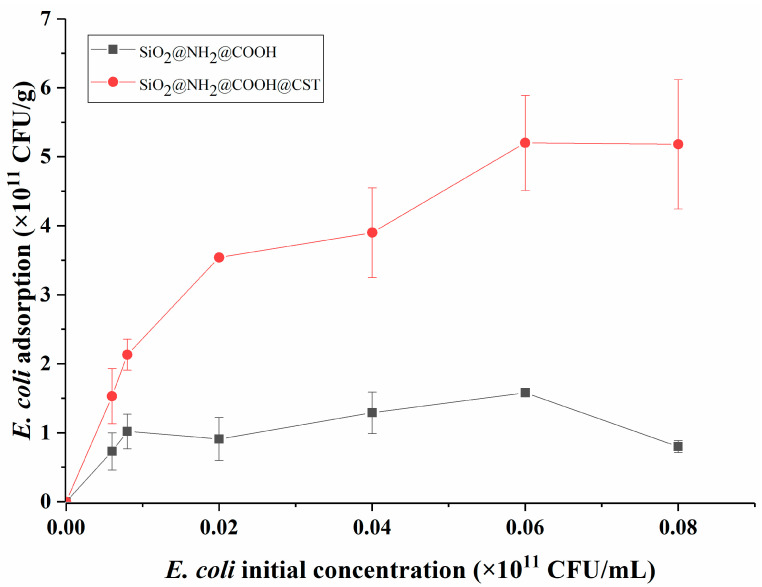
Adsorption isotherms of SiO_2_@NH_2_@COOH and SiO_2_@NH_2_@COOH@CST.

**Figure 5 molecules-27-08292-f005:**
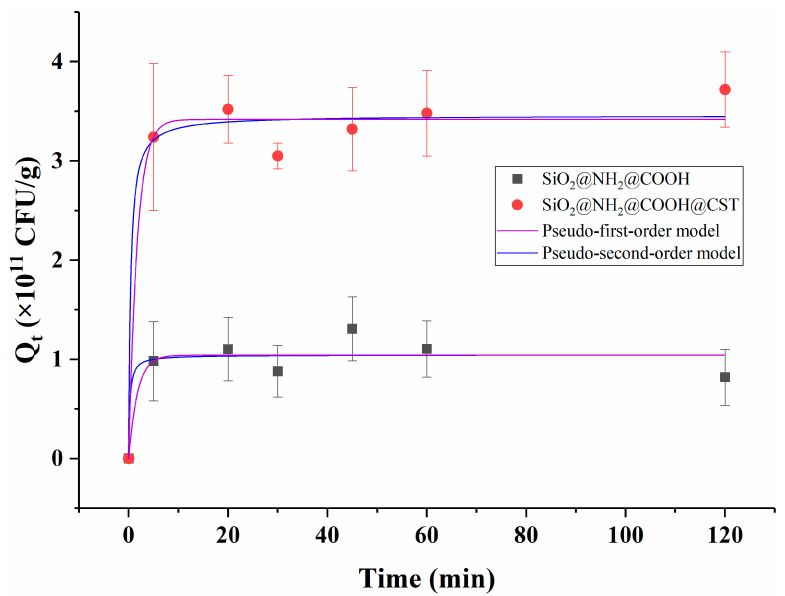
Pseudo-first-order models and pseudo-second-order models of *E*. *coli* on SiO_2_@NH_2_@COOH and SiO_2_@NH_2_@COOH@CST.

**Figure 6 molecules-27-08292-f006:**
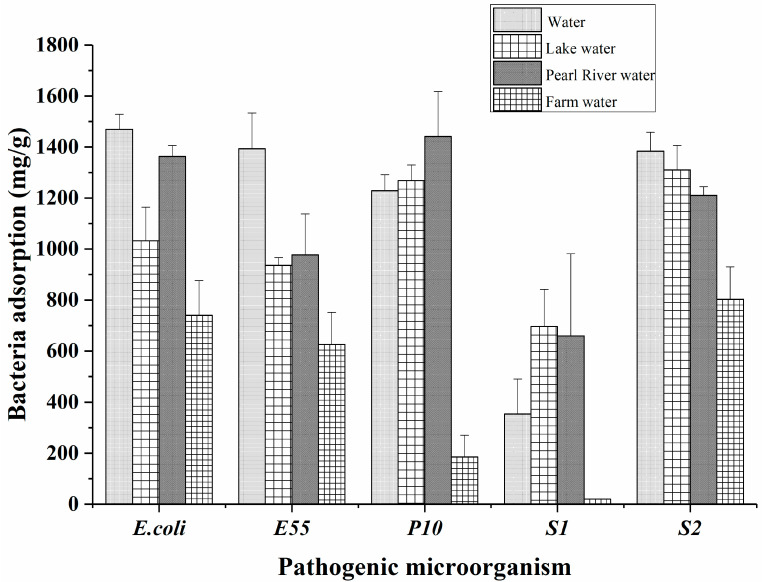
Adsorption of different bacteria by SiO_2_@NH_2_@COOH@CST in tap water, lake water, Pearl River water, and farm water. *E55*, *P10*, *S1*, and *S2* represent *E*. *coli*, *Pseudomonas aeruginosa* (*P*. *aeruginosa*), *Salmonellae*, and *Staphylococcus aureus* (*S*. *aureus*).

## Data Availability

Not applicable.

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
