# Peer review of "Synthesis and Characterization of Colistin-Functionalized Silica Materials for Rapid Capture of Bacteria in Water"

_molecules, 2022, doi:10.3390/molecules27238292_

Round 1
Reviewer 1 Report
Synthesis and characterization of colistin-functionalized silica materials for rapid capture of bacteria in water
In this manuscript titled “Synthesis and characterization of colistin-functionalized silica materials for rapid capture of bacteria in water” the authors have synthesized a new colistin-functionalized silica gel (SiO2@NH2@COOH@CST) material after carboxylation in the surface of the amino-modified silica. Types of linkers, linking means, reaction buffers and particle size of carriers were considered and characterization studies were conducted. The sorbent posed good and fast adsorption of gram negative bacteria and it can be used for bacteria detection and isolation.
A slight distinction about gram positive and gram negative bacteria may be added in the introduction part as it would be useful for readers.
Comments to the authors
The authors have used the acronym PMX-F in Line 54, Kindly expand it. Similarly expand SA in Line No. 111i.e in its first occurrence (but included in Line 330).
The adsorption capacity was found and reported to be low. Kindly add some literature to prove it. In Figure 1, the authors have used numerals 1- 5 to represent X-axis. It would be better to include the name for better understanding than numerals.
In section 2.2.2 FT-IR analysis, it was mentioned that FT-IR spectra of the prepared SiO2@NH2, 186 SiO2@NH2@COOH and SiO2@NH2@COOH@CST were tested and peaks were found at some places like 2964 cm-1, 2938 cm-1, 1699 cm-1, 1654 cm-1 and 1559 cm-1. Authors have used Bruker Vertex 70 infrared spectrometer (Bruker, Ger- 367 many), recording from 4000 cm-1 to 500 cm-1.But no Spectrum was included here for comparison. Kindly include an FTIR spectrum of the study here.
Many of the chemicals used as disinfectants are carcinogenic as they produce by products that are harmful. How does the authors justify that the currently synthesized colistin-functionalized silica materials is safe?
The authors have referred and cited some literature related to this study. More recently published literature if available can be added as some of the references cited in this research are quite old. Or is this quite novel than other studies?
Will this colistin-functionalized silica material has any significant effect on gram positive bacteria?
Does this colistin-functionalized silica material adsorb any other germs or micro-organisms that affect humans?
Slight English polishing is suggested for improved readability and flow.
Reviewer 2 Report
In their work Qiu and co-authors functionalize commercial silica spheres with colistin in order to obtain a material for the selective removal of bacteria from water. Different functionalization procedures and parameters are investigated and compared.
On the overall, the paper is clear and interesting; I just have minor comments.
- Line 11: it is not clear that SA means succinic anhydride, since this is explained only at the end of the paper (in the materials and methods section). Please, explain every acronym the first time you use it.
- Line 123: “…ratio of CST when IPTS and HDMI ARE used as linking agents” (ARE is missing)
- Figure 1: it may be clearer if for the samples (x-axis) you use short names instead of numbers (e.g., “no CTS” instead of “1”, “APTES” instead of 2, “APTES+6-AA” instead of 3, “IPTS” instead of 4, “HDMI” instead of 5)
- Line 134 and line 141: define the acronyms of MES and EDC
- Line 138-139: why the synthesis is more effective in organic buffer? Is there and explanation in the literature or do the authors have a hypothesis?
- Line 153: “the bacteria ADHESION to the larger…surface WAS less” or “the bacteria EDHERED to the larger…surface WERE less”. In general, check again English grammar throughout the text.
- Line 171: “…were all regular SPHERES with smooth surface”
- Line 172: “…and relatively DISPERSED/DISPERSION”
- Line 174: I suggest replacing “great” with “significant”
- Line 174-175: “The morphology of THE three kinds of silica...THE three kinds of silica gels were all dispersed and HAD a smooth surface”
- Line 206: please write all the mass loss with the same significant digits. By the way I doubt that the centesimal digit is significant.
- Line 213: “…was QUITE different”
- Line 222: What do the authors mean by “Adsorption isotherms3? Why “isotherms”? Please explain it better in the materials and methods section
- Line 233: “…which was conducive to capturing E.coli”: what does this mean? Please rephrase it
- Line 246-249: what’s the time of adsorption of commercial products (if there is any)? Is it comparable to that of the proposed solution?
- Line 273: replace “existence” with “presence”
- Line 279: “Factors AFFECTING bacterial adsorption”
- Line 290-291: what is the pH of bacterial environment in water normally? What is the pH of the samples the authors tested (lake water, farm water, etc.)?
- Line 315: replace “existed” with “exhibited”
- Line 316: why is adsorption in farm water lower? The authors should add an explanation or, at least, their hypotheses.
- Line 319: “It might be related…”: what does this sentence refer to? It is not clear; please rephrase it
- Line 386-388: the purpose of the addition of sucrose is not clear; please rephrase it.
